# Successful Pregnancy and Delivery at Term Following Intravenous Immunoglobulin Therapy with Heparin for Unexplained Recurrent Pregnancy Loss Suspected of Immunological Abnormalities: A Case Report and Brief Literature Review

**DOI:** 10.3390/jcm12041250

**Published:** 2023-02-04

**Authors:** Junichiro Mitsui, Kuniaki Ota, Yuko Takayanagi, Yurie Nako, Makiko Tajima, Atsushi Fukui, Kiyotaka Kawai

**Affiliations:** 1Graduate School of Medical and Dental Sciences, Comprehensive Reproductive Medicine, Tokyo Medical and Dental University, Tokyo 113-8510, Japan; 2Kameda IVF Clinic Makuhari, Reproductive Medicine, Chiba 261-0023, Japan; 3Fukushima Medical Center for Children and Women, Fukushima Medical University, Fukushima 960-1295, Japan; 4Department of obstetrics and Gynecology, Tokyo Rosai Hospital, Tokyo 143-0013, Japan; 5Department of Obstetrics and Gynecology, Hyogo Medical University, Nishinomiya 663-8131, Japan

**Keywords:** intravenous immunoglobulin, heparin, recurrent pregnancy loss, unexplained risk factor, immunotherapy

## Abstract

About 60% of cases of recurrent pregnancy loss have unexplained etiology. Immunotherapy for unexplained recurrent pregnancy loss is still unestablished. A 36-year-old woman, not obese, had a stillbirth at 22 gestational weeks and a spontaneous abortion at 8 weeks. She had been examined for recurrent pregnancy loss at previous clinics with no significant findings. When she visited our clinic, a hematologic test showed a Th1/Th2 ratio imbalance. Ultrasonography, hysteroscopy, and semen analysis showed no abnormalities. She successfully conceived by embryo transfer in hormone replacement therapy cycle. However, she had a miscarriage at 19 weeks. The baby had no deformities, but a chromosomal test was not performed, according to the parents’ will. The placenta pathologically suggested hemoperfusion problems. Her and her husband’s chromosomal tests showed normal karyotypes. Other examinations revealed a repeated Th1/Th2 ratio imbalance and a high resistance index of uterine radial artery blood flow. She was administered low-dose aspirin, intravenous immunoglobulin, and unfractionated heparin after the second embryo was transferred. Her baby was healthily born by cesarean section at 40 weeks. Intravenous immunoglobulin therapy can be a choice for recurrent miscarriage without risk factors because it has clinically beneficial influences on the patient’s immunological aberration.

## 1. Introduction

One to two percent of women and two to five percent of couples suffer from recurrent pregnancy loss (RPL) [1]. Etiologies include genetic aberrations, maternal autoantibodies, endocrine dysfunctions, and uterine abnormalities [2]. So far, over 50% of RPL cases are unexplained [3], but it is believed that it remains unexplained in 25% of patients, except for those with chromosomal abnormalities in embryos [4]. Recently, several studies have pointed out close associations between RPL and immunopathology [5,6,7]. Given the increasing evidence of the relationships, some immunomodulatory therapies have been evaluated in RPL patients [8,9,10].

As immunotherapy, intravenous immunoglobulin (IVIG) treatment has shown multiple functions: inhibition of the pathological activity of disease-associated autoantibodies [11], downregulation of Natural killer cell cytotoxicity [12], and suppression of helper 1 T cell (Th1) cytokines [13].

Despite the lack of evidence of improved pregnancy outcomes [14], it has been shown that starting IVIG administration before conception improves the live birth rate [15]. Nonetheless, the dosage and duration of IVIG administration are controversial for the treatment of RPL patients [15].

In the present study, we report a patient with a history of two mid-term miscarriages and one early miscarriage. She was diagnosed with pregnancy losses due to an immunological abnormality and finally delivered a healthy baby with intravenous immunoglobulin from hCG-positive diagnosis to 36 weeks of pregnancy. Additionally, we reviewed the related literature in this field.

## 2. Case Presentation

A 36-year-old woman, gravida 2 and para 1, visited our clinic with a desire to bear children. She had previously had a stillbirth at 22 gestational weeks and a spontaneous abortion at 8 gestational weeks. She had been examined for RPL at other clinics with no significant findings, and in particular, the patient’s laboratory test for APS (antiphospholipid syndrome) was performed. The results showed that LAC (lupus anticoagulant), aCL (anti-cardiolipin) antibodies, and aβ2GP1 (anti-β2 glycoprotein I) antibodies were all within reference values. She failed to conceive after one cycle of timed intercourse and one cycle of intrauterine insemination. No cultivatable oocytes were picked up in one cycle of controlled ovarian hyperstimulation (COH). Her body mass index was 24.5 kg/m^2^, and she had regular menstrual cycles. She had quit smoking for 6 years and never drank alcohol. The hematologic test did not show antiphospholipid antibody syndrome but showed an imbalance of the peripheral helper 1 T cell/helper 2 T cell (Th1/Th2) ratio (20.5) and a high value of antinuclear antibody (27.4-fold). To identify other immunological markers, the level of peripheral blood NK cell activity and endometrial CD16+/CD56dim cell ratio showed 26.0% and 5.01%, respectively, thus evaluated as normal. Ultrasonography showed no fibroids or adenomyosis, and hysteroscope showed no intrauterine deformation. The semen analysis and DNA fragmentation index were within normal range.

By COH with gonadotropin-releasing hormone antagonist, 23 oocytes were picked up, and 3 embryos were vitrified. She was confirmed with serum hCG elevation and successfully conceived by frozen embryo transfer in a hormone replacement therapy cycle. She was also treated with low-dose aspirin (LDA) (Bayaspirin®, Bayer Pharma AG, Osaka, Japan) and tacrolimus following implantation, due to suspicion of thrombophilia and the high peripheral Th1/Th2 ratio imbalance (16.8). However, despite LDA and tacrolimus, she had a miscarriage at 19 gestational weeks. The baby had no malformations, but a chromosomal test was not performed, according to the parents’ will. The cross-section of the placenta was pallor and presented with partial calcification. This suggested maternal disorder of blood flow to some extent. Her and her husband’s chromosomal test had normal karyotypes. However, they did not undergo preimplantation genetic testing for aneuploidy.

They were referred to another hospital to obtain a second opinion for investigating the cause of RPL. Examinations revealed a repeated peripheral Th1/Th2 ratio imbalance (17.1) and a relatively high resistance index of uterine radial artery blood flow (0.71). Considering these results, we needed to manage her immunological aberration and hemoperfusion disorder. Therefore, we administered approximately 400 mg/kg of intravenous immunoglobulin (IVIG) (Kenketsu Venoglobulin IH®, the Japan Blood Products Organization, Tokyo, Japan) every 4 weeks and 10,000 U of unfractionated heparin (UFH) (Heparin Calcium, Mochida Pharmaceutical Co., Ltd., Tokyo, Japan) a day, after she underwent second frozen embryo transfer (Figure 1). The pregnancy progress was uneventful; LDA was used until 28 weeks, IVIG until 35 weeks, and UFH until 36 weeks. Finally, she had a cesarean section at 40 gestational weeks; she had a female newborn weighing 2865 g. Both were discharged 7 days after birth.

## 3. Discussion

In this case, a woman with a history of recurrent miscarriages and stillbirth had no significant genetical or endocrinological abnormalities. Considering the results of blood and chromosomal tests, she was diagnosed with unexplained RPL (uRPL). However, due to the histopathological examination of the placenta at stillbirth and repeated blood tests, we suspected the RPL was affected by hemoperfusion disorder and immunological abnormalities. Finally, she was administered hybrid therapy, including LDA, UFH, and IVIG, and she conceived and delivered a live baby.

Immunomodulatory therapy for RPL has no evidence since most studies are observational studies. In the former studies, medium-dose IVIG was administered; the dosage was 20−50 g several times before or after pregnancy was confirmed [16,17,18]. A recent randomized controlled trial revealed that high-dose IVIG treatment (100 g) for RPL can improve pregnancy outcomes, including ongoing pregnancy and live birth rates [19]. In addition, Christiansen et al. described that high-dose IVIG effectively prevented RPL in a randomized, double-blind, placebo-controlled trial [20]. However, high-dose IVIG caused a high rate of preterm delivery and fetal growth restriction in women with uRPL [21]. Hence, we utilized medium-dose IVIG (400 mg/kg) until 36 gestational weeks and succeeded in pregnancy and delivery without perinatal adverse effects. It is unknown whether high or medium-dose IVIG is more effective as a treatment for RPL [22].

Treatment for uRPL includes anticoagulants, and their efficacy has been controversial [1,23,24]. LDA or heparin is widely used clinically as a single agent or in combination. A recent meta-analysis revealed that aspirin alone was associated with a lower live birth rate compared to heparin plus aspirin in RPL with antiphospholipid syndrome [25]. In this case, we utilized a combination of LDA and UFH, although the former frozen embryo transfer failed with LDA alone. Anticoagulation for uRPL should be a personalized treatment strategy, since uRPL is not a uniform group.

IVIG has long been an effective therapy for immunological disorders: primary immunodeficiency or idiopathic thrombocytopenic purpura, chronic inflammatory demyelinating polyneuropathy [26]. Such as this case, an imbalance of the Th1/Th2 ratio has been reported the etiology of uRPL [27,28]. In particular, IVIG may improve pregnancy outcomes in women with RPL because IVIG reduced Th1 cell levels, transcription factor expression, and type 1 cytokine levels, while increasing Th2 cell levels with the substantial decrease in the Th1/Th2 cell ratios [29]. In this case, we added IVIG to the previous treatment, which averted early miscarriage and led to delivery without causing a mid-term miscarriage after long-term administration. We supposed one of the possibilities that IVIG might modify the Th1/Th2 ratio imbalance as the immunomodulators and result in successful pregnancy and delivery although there is generally still no consensus on the effects of IVIG for RPL. To our knowledge, there are no studies in intervention trials using IVIG for uRPL while monitoring the Th1/Th2 balance till the late term of pregnancy. We need further study to confirm whether the intervention of IVIG modifies the Th1/Th2 ratio to immunologically maintain the pregnancy process. 

On the other hand, it has long been questioned whether immune cells in the peripheral blood are identical to immune cells in the endometrium since immune cells change phenotype depending on where they are present [30]. In other words, it is debatable whether Th cells in the endometrium reflect Th cells in the peripheral blood and can serve as clinical markers. Concretely, some researchers reported that peripheral and endometrial Th cells were similar in the distribution in uRP [31], although other researchers described that the proportions and percentage of Th cells were different across the endometrium and peripheral in uRPL [30,32]. We monitored peripheral Th cells and finally decided to use IVIG with UFH in this case, since IVIG was expected to modulate the immune environment at the interface of implantation. However, the global peripheral and endometrial immune landscape in RPL patients is still unclear. Therefore, to identify the accurate intervention with IVIF for uRPL, it is further necessary to develop a reliable strategy to characterize the peripheral blood and endometrial immune cell populations in uRPL to predict and evaluate the process of implantation and ongoing pregnancy.

Our patient had an abnormal peripheral T-cell profile and impaired blood perfusion. IVIG was used for the former, and LDA and UFH for the latter, to obtain a live birth. We believe immunological therapy, such as immunoglobulin, has enormous potential for treating RPL and needs further studies. A multimodal approach is often taken in uRPL, with various IVIG doses and duration, and combinations of medications. We hope a large-scale multicenter randomized prospective study with a consistent protocol will be conducted to consider our combination or dosage and duration in this study. 

## Figures and Tables

**Figure 1 jcm-12-01250-f001:**
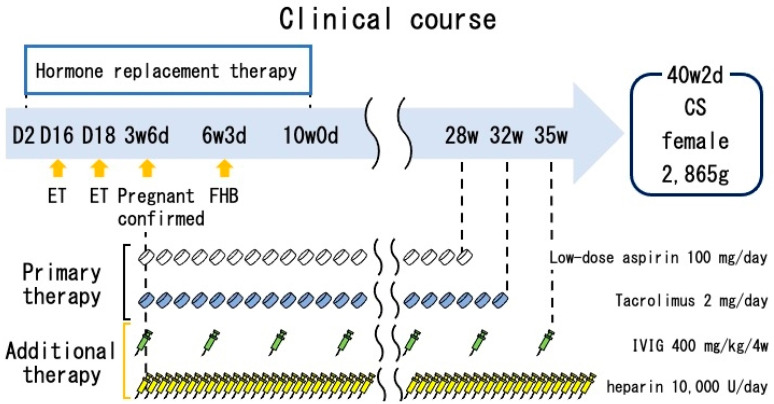
The successful clinical course of this case at the second embryo transfer. At the first embryo transfer, she was administered with LDA and tacrolimus. Considering additional results of examinations, at the second embryo transfer, she was administered with IVIG and UFH added to primary therapy. CS: cesarean section, ET: embryo transfer, FHB: fetal heartbeat.

## Data Availability

Not applicable.

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
