# Peer review of "Successful Pregnancy and Delivery at Term Following Intravenous Immunoglobulin Therapy with Heparin for Unexplained Recurrent Pregnancy Loss Suspected of Immunological Abnormalities: A Case Report and Brief Literature Review"

_jcm, 2023, doi:10.3390/jcm12041250_

Round 1

Reviewer 1 Report

This case report impressively shows the benefits of IVIG therapy for unexplained recurrent pregnancy loss. The case is clear and well presented. I notice the following points:

1) Were other causes of the abortions clarified in the patient's medical history? For example, has an antiphospholipid syndrome been ruled out as a cause in the patient?

2) Is there a possible connection and an explanation for the effect of the IVIG?

3) Was the Th1/Th2 ratio measured again after IVIG therapy to determine a possible effect of IVIG on the imbalance?

Author Response

Dear Editor and Reviewer.1

Thank you very much for reviewing our manuscript and offering valuable advice.

We have addressed your comments with point-by-point responses and revised the manuscript accordingly.

Reviewer1:

1) Were other causes of the abortions clarified in the patient's medical history? For example, has an antiphospholipid syndrome been ruled out as a cause in the patient?

Response:

We appreciate for your comments and agree with you.

Therefore, we added one sentence below as yellow-highlighted in page 2, lines 60-64.

She had been examined at other clinics for RPL with no significant findings, and especially the patient's laboratory test for APS (antiphospholipid syndrome) were performed. The results showed that LAC (lupus anticoagulant), aCL (anti-cardiolipin) antibodies, and aβ2GP1 (anti-β2 glycoprotein I) antibodies were all within reference values.

2) Is there a possible connection and an explanation for the effect of the IVIG?

Response:

We appreciate for your comments and agree with you.

Therefore, we added one sentence below as yellow-highlighted in page3-4, lines 133-140.

Especially, IVIG may improve pregnancy outcomes in women with RPL because IVIG reduced Th1 cell levels, transcription factor expression, and type 1 cytokine levels, while increasing Th2 cell levels with the substantial decrease in Th1/Th2 cell ratios [29]. In this case, we added IVIG to the previous treatment, which averted early miscarriage and led to delivery without causing a mid-term miscarriage after long-term administration. We supposed one of the possibilities that IVIG might modify Th1/Th2 ratio imbalance as the immunomodulators and result in successful pregnancy and delivery although there is generally still no consensus on the effects of IVIG for RPL.

3) Was the Th1/Th2 ratio measured again after IVIG therapy to determine a possible effect of IVIG on the imbalance?

Response:

We appreciate for your comments and agree with you.

Unfortunately, we didn’t measure Th1/Th2 ratio following ongoing pregnancy after the intervention with IVIG, but we have included your point as a consideration for future study (page 4, lines 141–144). Thank you for the suggestion.

Therefore, we added one paragraph below as yellow-highlighted.

To our knowledge, there are no studies in intervention trials using IVIG for uRPL while monitoring Th1/Th2 balance till the late term of pregnancy. We need further study to confirm whether the intervention of IVIG modifies Th1/Th2 ratio to immunologically maintain the pregnancy process.

Again, thank you for giving us the opportunity to strengthen our manuscript with your valuable comments and queries. We have worked hard to incorporate your feedback and hope that these revisions persuade you to accept our submission.

Sincerely,

Kuniaki Ota

Fukushima Medical Center for Children and Women, Fukushima Medical University

960-1295

Phone: (+81)24-547-1111

FAX: (+81)24-547-1386

e-mail address: [email protected]

Reviewer 2 Report

Dear authors,

I was pleased to read your interesting case report of a patient with uRPL, for whom the implementation of a combined therapeutic approach with IVIG led to a successful term pregnancy. The manuscript is overall well written and pertinent in this field of reproductive immunology.

Still, I have some minor comments that I believe could improve the document, and which I leave to your attention below.

1 – I understand the relevance of addressing Th1/Th2 imbalance along pregnancy, but was this the only immune assessment performed? For instance, were Tregs and/or Th17 also addressed? I believe the manuscript could be improved if a more comprehensive characterization of the patient is presented, for instance, in as a table or as supplementary data.

2 – Moreover, if you believe that your therapeutic intervention could directly interfere with the reported Th1/Th2 imbalance, did you monitor these along gestation? Can you comment on this, and if this has been performed, please add these data in the manuscript.

3 – How does peripheral Th imbalance relates to local Th populations at the uterine level? Authors should consider including this in discussion, relating to recent studies addressing these and other immune cell populations in local and systemic settings (i.e., doi.org/10.3389/fimmu.2022.994240, and/or https://doi.org/10.1038/s41598-017-03191-0).

4 – Please consider reviewing the first sentence of discussion, to bring all verbal forms into the past, as you continue using those afterwards in the text.

Kind regards. 

Author Response

Dear Editor and Reviewer.2

Thank you very much for reviewing our manuscript and offering valuable advice.

We have addressed your comments with point-by-point responses and revised the manuscript accordingly.

Reviewer2:

1 – I understand the relevance of addressing Th1/Th2 imbalance along pregnancy, but was this the only immune assessment performed? For instance, were Tregs and/or Th17 also addressed? I believe the manuscript could be improved if a more comprehensive characterization of the patient is presented, for instance, in as a table or as supplementary data.

Response:

We appreciate for your comments and agree with you.

Therefore, we added one sentence below as yellow-highlighted in page2, lines 70-72.

To identify other immunological markers, the level of peripheral blood NK cell activity and endometrial CD16+/CD56dim cell ratio respectively showed 26.0% and 5.01% to evaluate as normal.

2 – Moreover, if you believe that your therapeutic intervention could directly interfere with the reported Th1/Th2 imbalance, did you monitor these along gestation? Can you comment on this, and if this has been performed, please add these data in the manuscript.

Response:

We appreciate for your comments and agree with you.

This comment might be similar with reviewer 1 -3).

Unfortunately, we didn’t measure Th1/Th2 ratio following ongoing pregnancy after the intervention with IVIG, but we have included your point as a consideration for future study (page 4, lines 141–144). Thank you for the suggestion.

Therefore, we added one paragraph below as yellow-highlighted.

To our knowledge, there are no studies in intervention trials using IVIG for uRPL while monitoring Th1/Th2 balance till the late term of pregnancy. We need further study to confirm whether the intervention of IVIG modifies Th1/Th2 ratio to immunologically maintain the pregnancy process.

3 – How does peripheral Th imbalance relates to local Th populations at the uterine level? Authors should consider including this in discussion, relating to recent studies addressing these and other immune cell populations in local and systemic settings (i.e., doi.org/10.3389/fimmu.2022.994240, and/or https://doi.org/10.1038/s41598-017-03191-0).

Response:

We appreciate for your comments and agree with you.

Therefore, we added one paragraph below as yellow-highlighted in page 4, lines 145-158.

On the other hand, it has long been questioned whether immune cells in the peripheral blood are identical to immune cells in the endometrium since immune cells change phenotype depending on where they are present [30]. In other words, it is debatable whether Th cells in the endometrium reflect Th cells in the peripheral blood and can serve as clinical markers. Concretely, some researchers reported that peripheral and endometrial Th cells were similar in the distribution in uRP [31], although other researchers described that the proportions and percentage of Th cells were different across the endometrium and peripheral in uRPL [30,32]. We monitored peripheral Th cells and finally decided to use IVIG with UFH in this case since IVIG was expected to modulate the immune environment at the interface of implantation. However, the global peripheral and endometrial immune landscape in RPL patients is still unclear. Therefore, to identify the accurate intervention with IVIF for uRPL, it is further need to develop a reliable strategy to characterize the peripheral blood and endometrial immune cell populations in uRPL to predict and evaluate the process of implantation and ongoing pregnancy.

4 – Please consider reviewing the first sentence of discussion, to bring all verbal forms into the past, as you continue using those afterwards in the text.

Response:

We appreciate for your comments and agree with you.

Therefore, we modified under your instruction as yellow-highlighted in page3, lines 105.

Again, thank you for giving us the opportunity to strengthen our manuscript with your valuable comments and queries. We have worked hard to incorporate your feedback and hope that these revisions persuade you to accept our submission.

Sincerely,

Kuniaki Ota

Fukushima Medical Center for Children and Women, Fukushima Medical University

960-1295

Phone: (+81)24-547-1111

FAX: (+81)24-547-1386

e-mail address: [email protected]
